# Pre-training Intent-Aware Encoders for Zero- and Few-Shot Intent Classification

**Mujeen Sung**[1*]  **James Gung**[2]  **Elman Mansimov**[2]  **Nikolaos Pappas**[2]
**Raphael Shu**[2]  **Salvatore Romeo**[2]  **Yi Zhang**[2]  **Vittorio Castelli**[2]
Korea University[1]    AWS AI Labs[2]
mujeensung@korea.ac.kr
{gungj,mansimov,nppappa}@amazon.com
{zhongzhu,romeosr,yizhngn,vittorca}@amazon.com

## Abstract

Intent classification (IC) plays an important role in task-oriented dialogue systems. However, IC models often generalize poorly when training without sufficient annotated examples for each user intent. We propose a novel pre-training method for text encoders that uses contrastive learning with intent psuedo-labels to produce embeddings that are well-suited for IC tasks, reducing the need for manual annotations. By applying this pre-training strategy, we also introduce **P**re-trained **I**ntent-aware **E**ncoder (PIE), which is designed to align encodings of utterances with their intent names. Specifically, we first train a tagger to identify key phrases within utterances that are crucial for interpreting intents. We then use these extracted phrases to create examples for pre-training a text encoder in a contrastive manner. As a result, our PIE model achieves up to 5.4% and 4.0% higher accuracy than the previous state-of-the-art text encoder for the N-way zero- and one-shot settings on four IC datasets.

## 1 Introduction

Identification of user intentions, a problem known as intent classification (IC), plays an important role in task-oriented dialogue (TOD) systems. However, it is challenging for TOD developers to collect data and re-train models when designing new intent classes. Recent studies have aimed to tackle this challenge by applying zero- and few-shot text classification methods and leveraging the semantics of intent label names (Liu et al., 2019a; Krone et al., 2020; Burnyshev et al., 2021; Mueller et al., 2022; Zhang et al., 2022; Lamanov et al., 2022; Liu et al., 2022) .

Dopierre et al. (2021a) compare various classification methods on few-shot IC tasks and find that Prototypical Networks (Snell et al., 2017) consistently show strong performance when combined

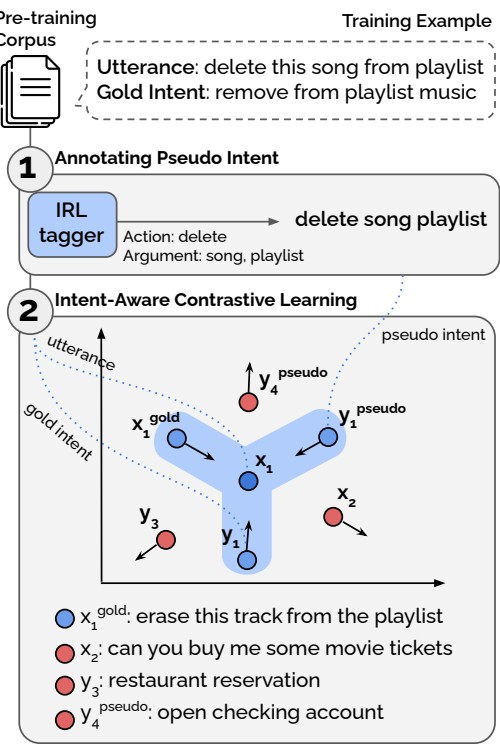

Figure 1: Overview of pre-training the intent-aware encoder (PIE). Given an utterance $x_1$ from pre-training corpus, we generate a pseudo intent name $y_1^{\text{pseudo}}$ using labels from the intent role labeling (IRL) tagger. Our PIE model is then optimized by pulling the gold utterance $x_1^{\text{gold}}$, gold intent $y_1$, and pseudo intent $y_1^{\text{pseudo}}$ close to the input utterance $x_1$ in the embedding space.

with transformer-based text encoders. Prototypical Networks use text encoders to construct class representations and retrieve correct classes given queries based on a similarity metric. Dopierre et al. (2021a) also stress that few-shot learning techniques and text encoders can have an orthogonal impact on classification performance. Thus, although some studies have focused on improving learning techniques for few-shot IC tasks (Dopierre et al., 2021b; Chen et al., 2022), better text encoder selection should also be considered as an impor-

---

*Work performed during an internship at AWS AI Labs.

tant research direction. Ma et al. (2022) observe that sentence encoders pre-trained on paraphrase or natural language inference datasets serve as strong text encoders for Prototypical Networks. However, existing sentence encoders are not explicitly designed to produce representations for utterances that are similar to their intent names. Therefore, their abilities are limited in zero- and few-shot settings where predictions may heavily rely on the semantics of intent names. Pre-training encoders to align user utterances with intent names can mitigate this issue; however, it is typically expensive to obtain annotations for a diverse intent set.

In this paper, we propose a novel pre-training method for zero- and few-shot IC tasks (Figure 1). Specifically, we adopt intent role labeling (IRL) (Zeng et al., 2021), which is an approach for identifying and assigning roles to words or phrases that are relevant to user intents in sentences. Once we obtain the IRL predictions, we convert them to the pseudo intent names of query utterances and use them to pre-train the encoder in a contrastive learning fashion. This intent-aware contrastive learning aims to not only align utterances with their pseudo intent names in the semantic embedding space, but also to encourage the encoder to pay attention to the intent-relevant spans that are important for distinguishing intents. To the best of our knowledge, this work is the first to extract key information from utterances and use it as pseudo labels for pre-training intent-aware text encoders.

The contributions of our work are as follows:

- First, we propose an algorithm for generating pseudo intent names from utterances across several dialogue datasets and publicly release the associated datasets.

- Second, by applying intent-aware contrastive learning on gold and pseudo intent names, we build **P**re-trained **I**ntent-aware **E**ncoder (PIE), which is designed to align encodings of utterances with their intent names.

- Finally, experiments on four IC datasets demonstrate that the proposed model outperforms the state-of-the-art work (Dopierre et al., 2021b; Ma et al., 2022) by up to 5.4% and 4.0% on N-way zero- and one-shot settings, respectively.

## 2 Background: Prototypical Networks for Intent Classification

Prototypical Networks (Snell et al., 2017) is a meta-learning approach that enables classifiers to quickly adapt to unseen classes when only a few labeled examples are available. Several studies have demonstrated the effectiveness of Prototypical Networks when building intent classifiers with a few example utterances (Krone et al., 2020; Dopierre et al., 2021a; Chen et al., 2022). They first define a few-shot IC task, also known as an *episode* in the meta-learning context, with K example utterances from N intent classes (i.e., K×N utterances in a single episode). At the training time, the intent classifiers are optimized on a series of these episodes. Example utterances for each intent class are called a *support set*, and are encoded and averaged to produce a class representation, called a *prototype*. This can be formulated as follows:

$$\mathbf{c}_n = \frac{1}{K} \sum_{x_{n,i} \in S_n} f_\phi(x_{n,i}) \quad (1)$$

where $S_n$ denotes the support set of the n-th intent class, $x_{n,i}$ denotes the i-th labeled example of the support set $S_n$, $f_\phi(\cdot)$ denotes a trainable encoder, and $\mathbf{c}_n$ denotes the n-th prototype. At the inference time, the task is to map the query utterance representation to the closest prototype in a metric space (e.g., Euclidean) among the N prototypes. When there are N intent classes and each intent class has K example utterances, this setting is called N-way K-shot intent classification.

Ma et al. (2022) suggest that leveraging intent names as additional support examples is beneficial in few-shot IC tasks because the semantics of intent names can give additional hints to example utterances. When intents are used as additional support examples, the new prototype representations can be formulated as follows:

$$\mathbf{c}_n^{\text{label}} = \frac{1}{K+1} [[ \sum_{x_{n,i} \in S_n} f_\phi(x_{n,i})] + f_\phi(y_n)] \quad (2)$$

where $y_n$ is the intent name of the utterance in the n-th support set, and $\mathbf{c}_n^{\text{label}}$ is the n-th prototype using intent names as support. By using intents as support examples, it is possible to classify input utterances without example utterances in a zero-shot fashion. Specifically, the prototypes in Equation (2) can be calculated as $\mathbf{c}_n^{\text{label}} = f_\phi(y_n)$ based solely on intent names, which facilitates the zero-shot IC.

## 3 Pseudo Intent Name Generation

To pre-train an encoder $f_\phi$ that works robustly in zero- or few-shot IC settings, a variety of predefined intent names are required. Because annotating them is expensive, we opt to automatically generate *pseudo* intent names from utterances in our pre-training data. To annotate pseudo intents, we employ a tagging method, intent role labeling (IRL). IRL can be considered similar to semantic role labeling (SRL), which is a task of assigning general semantic roles to words or phrases in sentences (Palmer et al., 2010). However, IRL focuses on providing an extractive summary of the intent expressed in a user's utterance, annotating important roles with respect to the goal of the user rather than a predicate. Specifically, it tags words or phrases that are key to interpret intent.

IRL was first introduced by Zeng et al. (2021) for discovering intents from utterances, but their tagger focuses only on Chinese utterances. In this section, we outline the process of building the IRL tagger from scratch. We provide a description of how we annotate IRL training data on English utterances (Section 3.1), the training procedure for the IRL tagger (Section 3.2), and the utilization of IRL predictions for generating pseudo intent names to pre-train our model (Section 3.3).

### 3.1 Annotating Intent Roles

We define six intent role labels, *Action, Argument, Request, Query, Slot, and Problem*, for extracting intent-relevant spans from utterances. *Action* is a word or phrase (typically a verb or verb phrase) that describes the main action relevant to an intent in an utterance. *Argument* is an argument of an action, or entity/event that is important to interpreting an intent. *Request* indicates a request for something such as a question or information-seeking verb. *Query* indicates the expected type of answer to a question or request for information, or a requested entity to be obtained or searched for. *Slot* is an optional/variable value provided by the speaker that does not impact the interpretation of an intent. Finally, *Problem* describes some problematic states or events, and typically makes an implicit request.

Based on these definitions, we manually annotate IRL labels on a subset of utterances from SGD (Rastogi et al., 2020). Table 1 shows the statistics and examples of each IRL label from 3,879 utterances. To train and evaluate the IRL tagger, we split annotations into training, valida-

tion, and test sets with 3,121 / 379 / 379 utterances, respectively, which is approximately an 80:10:10 ratio.

| Label | Count | Example |
|---|---|---|
| Action | 2,163 | I want to **book**$_{ACT}$ a flight |
| Argument | 2,011 | I want to book a **flight**$_{ARG}$ |
| Request | 3,002 | Can you **show**$_{REQ}$ me my account balance |
| Query | 3,247 | Can you show me my **account balance**$_{QRY}$ |
| Slot | 2,030 | Can you show me my account balance for **my checking account**$_{SLT}$ |
| Problem | 45 | I'm starting to get **hungry**$_{PRB}$ |

Table 1: Statistics and examples of each IRL label from 3,879 utterances.

### 3.2 Training the IRL Tagger

Using manually curated IRL annotations, we formulate IRL as a sequence tagging problem. Specifically, we assign each token in an utterance with one of the 13 IRL labels under the Beginning-Inside–Outside (BIO) scheme (e.g., B-Action, I-Action, or O). The IRL model is, then, trained to predict the correct IRL labels of the tokens using the cross entropy loss. We use *RoBERTa-base* (Liu et al., 2019c) as the initial model for the IRL tagger. Table 2 shows the precision, recall, and F1 scores of each IRL label on the test set.

| Label | P (%) | R (%) | F1 (%) |
|---|---|---|---|
| Action | 89.3 | 86.9 | 88.1 |
| Argument | 85.8 | 85.4 | 85.6 |
| Request | 90.9 | 93.5 | 92.1 |
| Query | 92.2 | 95.8 | 94.0 |
| Slot | 82.8 | 84.6 | 83.7 |
| Problem | 35.6 | 41.7 | 38.1 |

Table 2: Precision, recall, and F1 scores of each IRL label on the test set.

### 3.3 Generating Pseudo Intents

After obtaining the IRL tagger, we leverage it to predict IRL labels for tokens in utterances from pre-training corpus described in Section 5.2. To generate pseudo intent names, we simply concatenate all spans that have been predicted as IRL labels in each utterance. Table 3 shows some examples of IRL predictions from utterances and corresponding pseudo intent names.

| Utterances | Pseudo Intents | Gold Intents |
|---|---|---|
| I'm calling because i'd like to **open**$_{\text{ACT}}$ an **account**$_{\text{ARG}}$ | open account | |
| I'd like to **open**$_{\text{ACT}}$ a **savings**$_{\text{SLT}}$ **account**$_{\text{ARG}}$ please | open savings account | open account |
| So I need to **sign up**$_{\text{ACT}}$ for a a **savings**$_{\text{SLT}}$ **account**$_{\text{ARG}}$ | sign up savings account | |
| Can you **buy**$_{\text{ACT}}$ me some **movie tickets**$_{\text{ARG}}$ | buy movie tickets | |
| I am looking to **book**$_{\text{ACT}}$ **movie tickets**$_{\text{ARG}}$ | book movie tickets | buy movie tickets |
| I am looking to **purchase**$_{\text{ACT}}$ **movie tickets**$_{\text{ARG}}$ | purchase movie tickets | |
| **Delete**$_{\text{ACT}}$ this **song**$_{\text{ARG}}$ from **playlist**$_{\text{ARG}}$ | delete song playlist | |
| **Erase**$_{\text{ACT}}$ this **track**$_{\text{ARG}}$ from the **playlist**$_{\text{ARG}}$ | erase track playlist | remove from playlist music |
| Could you **remove**$_{\text{ACT}}$ this **song**$_{\text{ARG}}$ permanently | remove song | |

Table 3: Some examples of IRL predictions (**boldfaced**) from utterances, extracted pseudo intent names, and gold intent names annotated in the original dataset.

## 4 Intent-Aware Contrastive Learning

We aim to build an encoder that produces similar representations between utterances and the corresponding intent names. In this section, we introduce the intent-aware contrastive learning approach using triples of an utterance, gold intent, and pseudo intent from various dialogue datasets.

Our training objective is designed to align the representations of utterances and their intent names in the semantic embedding space. For this purpose, we use the InfoNCE loss (van den Oord et al., 2018), which pulls positive pairs close to each other and pushes away negative pairs. The loss for the i-th sample $x_i$ is formulated as follows:

$$\ell(x_i, \boldsymbol{y}) = \frac{\exp\left[\,\text{sim}(f_\phi(x_i), f_\phi(y_i))\,\right]}{\sum_k^N \exp\left[\,\text{sim}(f_\phi(x_i), f_\phi(y_k))\,\right]}, \quad (3)$$

where $\boldsymbol{y} = \langle y_1, y_2, \ldots, y_N \rangle$ are pairs of the input $x_i$ with a batch size of $N$, and $\text{sim}(\cdot)$ denotes the cosine similarity between two embeddings. Again, $f_\phi(\cdot)$ denotes any text encoder that represents intent names or utterances in the embedding space. Note that pairs that are not positive in a batch are treated as negative pairs.

We here define three types of positive pairs, two of which are supervised and one is semi-supervised. The first of the supervised positive pairs is between the input utterances and their gold intent names annotated in the pre-training datasets. The equation used is as follows:

$$\mathcal{L}_{\text{gold\_intent}} = -\frac{1}{N}\sum_i^N \ell(x_i, \boldsymbol{y}^{\text{gold}}), \quad (4)$$

where $y_i^{\text{gold}}$ is the a gold intent name of $x_i$.

The second supervised positive pair is between the input utterances and their gold utterances. We define gold utterances as randomly sampled utterances that share the same gold intent names as the input utterances:

$$\mathcal{L}_{\text{gold\_utterance}} = -\frac{1}{N}\sum_i^N \ell(x_i, \boldsymbol{x}^{\text{gold}}), \quad (5)$$

where $x_i^{\text{gold}}$ is the gold utterance of $x_i$.

Finally, the semi-supervised positive pairs are between the input utterances and their pseudo intent names:

$$\mathcal{L}_{\text{pseudo}} = -\frac{1}{N}\sum_i^N \ell(x_i, \boldsymbol{y}^{\text{pseudo}}), \quad (6)$$

where $y_i^{\text{pseudo}}$ denotes the pseudo intent name of $x_i$ constructed by the IRL tagger, as described in Section 3.3

Our final loss is a combination of these three losses as follows:

$$\mathcal{L} = \mathcal{L}_{\text{gold\_intent}} + \mathcal{L}_{\text{gold\_utterance}} + \lambda \mathcal{L}_{\text{pseudo}}, \quad (7)$$

where $\lambda$ is the weight term of the semi-supervised loss term.

## 5 Experiment

### 5.1 Baselines

To evaluate the effectiveness of our proposed PIE model, we compare it with the following state-of-the-art approaches for few-shot IC tasks: ProtoNet (Dopierre et al., 2021a) and ProtAugment (Dopierre et al., 2021b) as fine-tuning methods, and SBERT$_{\text{Paraphrase}}$ (Ma et al., 2022) as a pre-trained text encoder.

ProtoNet is a meta-training approach that fine-tunes encoders by using a series of episodes constructed on task-specific training sets. ProtAugment is an advanced method derived from ProtoNet, which augments paraphrased utterances within episodes to mitigate overfitting caused by the biased distribution introduced by a limited number of training examples. The authors of ProtoNet and ProtAugment perform additional pre-training *BERT-base-cased* (110M) using training utterances and the language model objective, and use it as their initial model. We refer to this model as $\text{BERT}_{\text{TAPT}}$, inspired by task-adaptive pre-training (TAPT) (Gururangan et al., 2020).

$\text{SBERT}_{\text{Paraphrase}}$ is a text encoder pre-trained on large-scale paraphrase text pairs (Reimers and Gurevych, 2019). Ma et al. (2022) discover that this pre-trained text encoder can produce good utterance embeddings without any fine-tuning on task-specific datasets. Although the authors leverage $\text{SBERT}_{\text{Paraphrase}}$ solely at the inference stage of Prototypical Networks, we conduct additional experiments by fine-tuning the encoder using ProtoNet and ProtAugment as baselines. Note that we reproduce the performance of $\text{SBERT}_{\text{Paraphrase}}$ using *paraphrase-mpnet-base-v2*[1] (110M), which has the same number of parameters as *BERT-base-cased*, for a fair comparison.

## 5.2 Pre-training Datasets

| | Dataset | Utterances | Gold Intents | Pseudo Intents |
|---|---|---|---|---|
| **Train** | TOP (+v2) | 31,111 | 61 | 23,711 |
| | DSTC11-T2 | 4,459 | 148 | 3,304 |
| | SGD | 3,561 | 44 | 2,647 |
| | Total | 39,117 | 252 | 29,577 |
| **Val** | MultiWOZ 2.2 | 586 | 10 | - |

Table 4: Pre-training datasets for the PIE model.

We collect four dialogue datasets to pre-train and one to validate our encoder: TOP (Gupta et al., 2018)), TOPv2 (Chen et al., 2020), DSTC11-T2 (Gung et al., 2023), SGD (Rastogi et al., 2020), and MultiWOZ 2.2 (Zang et al., 2020). Dialogues in SGD and MultiWOZ 2.2 datasets consist of multi-turn utterances, and these utterances are often ambiguous when context around them is not given (e.g., 'Can you suggest something else?' is labeled as 'LookupMusic'). To minimize this am-

biguity, we use the first-turn utterance of each dialogue from these datasets. Furthermore, the number of utterances between intents in the raw datasets is highly imbalanced. To alleviate this imbalance, we set the maximum number of utterances per intent of the TOP and DSTC11-T2 datasets to 1000 and the SGD and MultiWOZ 2.2 datasets to 100. We then annotate the IRL labels on the utterances using the IRL tagger. Based on the IRL predictions, we filter utterances when no *Action*, *Argument*, or *Query* labels are detected, because they are likely to lack information for interpreting user intents. Finally, we treat MultiWOZ 2.2 as the validation set for tuning the hyperparameters of the pre-training stage. Table 4 summarizes the statistics of the datasets.

## 5.3 Downstream Datasets

| Dataset | Domains | Utterances | Intents | | |
|---|---|---|---|---|---|
| | | | Train | Valid | Test |
| Banking77 | 1 | 13,083 | 25 | 25 | 27 |
| HWU64 | 21 | 11,036 | 23 | 16 | 25 |
| Liu54 | 21 | 25,478 | 18 | 18 | 18 |
| Clinc150 | 10 | 22,500 | 50 | 50 | 50 |

Table 5: The statistics of four IC datasets.

We evaluate our PIE model and baseline models on four IC datasets (Table 5).

Banking77 (Casanueva et al., 2020) is an IC dataset in the banking domain. As there are many overlapping tokens between intent names (e.g. 'verify top up', 'top up limits', and 'pending top up'), fine-grained understanding is required when correctly classifying intents for this dataset. HWU64 (Liu et al., 2019b) is a dataset in 21 different domains, such as alarm and calendar for a home assistant robot. Liu54 (Liu et al., 2019b) is a dataset collected from Amazon Mechanical Turk, and workers designed utterances for given intents. Clinc150 (Larson et al., 2019) is a dataset that includes a wide range of intents from ten different domains such as 'small talk' and 'travel'.

| | Banking77 | HWU64 | Liu54 | Clinc150 |
|---|---|---|---|---|
| TOP (+v2) | 0 | 2 | 0 | 2 |
| DSTC11-T2 | 3 | 0 | 0 | 3 |
| SGD | 0 | 0 | 0 | 1 |
| **All** | 3/77 | 2/64 | 0/54 | 6/150 |

Table 6: Number of overlapping intent names between the pre-training data and downstream data.

[1]https://huggingface.co/sentence-transformers/paraphrase-mpnet-base-v2

| Method | Fine-tuning | Banking77 | | HWU64 | | Liu54 | | Clinc150 | | Average | |
|---|---|---|---|---|---|---|---|---|---|---|---|
| | | K=0 | K=1 | K=0 | K=1 | K=0 | K=1 | K=0 | K=1 | K=0 | K=1 |
| BERT$_{TAPT}$ | | - | $50.8_{\pm1.5}$ | - | $52.1_{\pm1.1}$ | - | $52.7_{\pm2.6}$ | - | $50.7_{\pm4.1}$ | - | 51.6 |
| SBERT$_{Paraphrase}$ | - | - | $83.6_{\pm1.8}$ | - | $82.2_{\pm1.0}$ | - | $82.3_{\pm2.2}$ | - | $94.7_{\pm0.6}$ | - | 85.7 |
| PIE (Ours) | | - | $\mathbf{86.1}_{\pm1.3}$ | - | $\mathbf{86.0}_{\pm1.7}$ | - | $\mathbf{85.6}_{\pm1.9}$ | - | $\mathbf{96.2}_{\pm0.5}$ | - | **88.5** |
| L-BERT$_{TAPT}$ | | $27.7_{\pm1.8}$ | $41.0_{\pm1.5}$ | $38.5_{\pm2.4}$ | $53.6_{\pm1.5}$ | $51.7_{\pm4.6}$ | $60.0_{\pm4.1}$ | $39.5_{\pm1.4}$ | $51.8_{\pm3.2}$ | 39.3 | 51.6 |
| L-SBERT$_{Paraphrase}$ | - | $86.9_{\pm1.9}$ | $90.9_{\pm0.6}$ | $83.6_{\pm1.8}$ | $89.0_{\pm1.4}$ | $79.9_{\pm3.8}$ | $89.8_{\pm1.9}$ | $94.3_{\pm1.1}$ | $97.7_{\pm0.4}$ | 86.2 | 91.9 |
| L-PIE (Ours) | | $\mathbf{88.3}_{\pm2.2}$ | $\mathbf{92.4}_{\pm0.7}$ | $\mathbf{87.7}_{\pm2.6}$ | $\mathbf{92.2}_{\pm1.5}$ | $\mathbf{83.8}_{\pm3.7}$ | $\mathbf{91.7}_{\pm1.4}$ | $\mathbf{96.5}_{\pm0.8}$ | $\mathbf{98.3}_{\pm0.4}$ | **89.1** | **93.7** |
| L-BERT$_{TAPT}$ | | $85.7_{\pm2.3}$ | $91.5_{\pm1.0}$ | $81.4_{\pm1.3}$ | $86.6_{\pm1.3}$ | $80.3_{\pm3.1}$ | $88.2_{\pm1.2}$ | $93.0_{\pm1.9}$ | $96.9_{\pm0.6}$ | 85.1 | 90.8 |
| L-SBERT$_{Paraphrase}$ | ProtoNet | $\mathbf{90.9}_{\pm1.9}$ | $\mathbf{94.5}_{\pm0.5}$ | $85.8_{\pm2.8}$ | $91.1_{\pm1.6}$ | $83.7_{\pm4.3}$ | $92.1_{\pm1.6}$ | $97.1_{\pm0.7}$ | $\mathbf{98.6}_{\pm0.2}$ | 89.4 | 94.0 |
| L-PIE (Ours) | | $90.7_{\pm2.2}$ | $94.3_{\pm0.7}$ | $\mathbf{86.6}_{\pm4.0}$ | $\mathbf{92.1}_{\pm1.7}$ | $\mathbf{85.0}_{\pm3.9}$ | $\mathbf{92.4}_{\pm1.5}$ | $\mathbf{97.3}_{\pm0.3}$ | $\mathbf{98.6}_{\pm0.3}$ | **89.9** | **94.4** |
| L-BERT$_{TAPT}$ | | $89.2_{\pm2.1}$ | $93.4_{\pm0.5}$ | $87.0_{\pm2.6}$ | $89.8_{\pm1.1}$ | $83.0_{\pm4.6}$ | $90.9_{\pm0.9}$ | $95.3_{\pm1.0}$ | $97.7_{\pm0.2}$ | 88.6 | 92.9 |
| L-SBERT$_{Paraphrase}$ | ProtAugment | $92.3_{\pm1.1}$ | $\mathbf{94.8}_{\pm0.4}$ | $87.3_{\pm2.5}$ | $91.7_{\pm1.6}$ | $84.1_{\pm3.3}$ | $92.5_{\pm1.4}$ | $97.0_{\pm0.7}$ | $98.5_{\pm0.2}$ | 90.2 | 94.4 |
| L-PIE (Ours) | | $\mathbf{92.4}_{\pm1.0}$ | $\mathbf{94.8}_{\pm0.4}$ | $\mathbf{88.8}_{\pm3.0}$ | $\mathbf{92.4}_{\pm1.6}$ | $\mathbf{86.0}_{\pm3.3}$ | $\mathbf{92.9}_{\pm1.3}$ | $\mathbf{97.6}_{\pm0.4}$ | $\mathbf{98.7}_{\pm0.2}$ | **91.2** | **94.7** |

Table 7: 5-way K-shot intent classification performance of pre-trained models with and without fine-tuning on four test sets. Averaged accuracies and standard deviations across five class splits are reported. The 'L-' prefixes indicate the use of intent label names when creating prototypes, enabling zero-shot evaluation. Highest scores are **boldfaced**.

Before proceeding, we examine the number of intent names that overlap between the pre-training data and downstream data (Table 6). When comparing the intent names, we first apply stemming ('restaurant reservation' → 'restaur reserve') and arrange the tokens in alphabetical order ('restaur reserv' → 'reserv restaur') for each intent name. This approach aims to maximize the recall of overlapping intent names. Consequently, we find that only 11 out of 345 intent names from the downstream data overlaps (e.g, 'reserve restaurant' intent in the pre-training data and 'restaurant reservation' intent in the downstream data).

## 5.4 Implementation Details

Here, we describe detailed information when pre-training our PIE model and employing it for zero- and few-shot IC tasks.

We use *paraphrase-mpnet-base-v2*, the same encoder used in the SBERT$_{Paraphrase}$ baseline, as an initial model for further pre-training in our approach. The hyperparameters are tuned based on the validation set described in Section 5.2. As a result, we set the training epochs to 1, the learning rate to 1e-6, the batch size to 50, and $\lambda$ to 2.

After pre-training, we apply the model to zero- and few-shot IC tasks in the 5-way and N-way settings. The baselines we compare with only experiment with a 5-way setting where the task is predicting the correct intent from among five candidate classes. We further include an N-way setting, where N can be much larger than five because, in practice, it is often required to assign more than five intents in building TOD systems. When evaluating models on the N-way setting, we use all the intent classes in the test set as candidate intents, for example, 27-way for Banking77 and 50-way for Clinc150. We set K, which is the number of examples per intent in an episode, to 0 and 1 to experiment with zero- and one-shot IC. Finally, we treat intent labels as examples when creating prototypes for each intent. This enables experiments in the zero-shot setting and enhances performance in the few-shot setting. To denote the usage of labels as examples, we append 'L-' prefixes to the method names (e.g., L-PIE).

## 5.5 Results

**5-way K-shot intent classification** Table 7 shows 5-way K-shot IC performance on four test sets. The results demonstrate that PIE achieves an average accuracy of 88.5%, surpassing SBERT$_{Paraphrase}$, which is considered the strongest baseline model, by 2.8% in the one-shot setting. This highlights the effectiveness of our pre-training strategy. Additionally, where intent labels are used as examples, our L-PIE model achieves 89.1% and 93.7% in the zero-shot and one-shot settings, respectively, consistently outperforming L-SBERT$_{Paraphrase}$ by 2.9% and 1.8%. It is worth noting that the L-PIE model also significantly outperforms L-BERT$_{TAPT}$ + ProtoNet, which fine-tunes an encoder on the target datasets, by a substantial margin of 4.0% and 2.9%. This shows that our proposed approach builds an effective intent classifier that performs well even prior to fine-tuning on task-specific data. Our L-PIE model shows further improvement when fine-tuned with ProtAugment, outperforming the strongest baseline L-SBERT$_{Paraphrase}$ + ProtAugment by 1.0% in zero-shot IC.

| Model | Fine-tuning | Banking77$_{N=27}$ | | HWU64$_{N=25}$ | | Liu54$_{N=18}$ | | Clinc150$_{N=50}$ | | Average | |
|---|---|---|---|---|---|---|---|---|---|---|---|
| | | K=0 | K=1 | K=0 | K=1 | K=0 | K=1 | K=0 | K=1 | K=0 | K=1 |
| BERT$_{TAPT}$ | | - | $28.4_{\pm1.8}$ | - | $28.4_{\pm1.5}$ | - | $32.1_{\pm4.1}$ | - | $26.3_{\pm0.6}$ | - | 28.8 |
| SBERT$_{Paraphrase}$ | - | - | $62.3_{\pm2.3}$ | - | $58.0_{\pm1.0}$ | - | $68.8_{\pm3.1}$ | - | $76.7_{\pm1.7}$ | - | 66.5 |
| PIE (Ours) | | - | $\mathbf{65.3}_{\pm2.9}$ | - | $\mathbf{66.0}_{\pm2.9}$ | - | $\mathbf{73.5}_{\pm2.8}$ | - | $\mathbf{82.5}_{\pm1.7}$ | - | **71.8** |
| L-BERT$_{TAPT}$ | | $9.0_{\pm1.4}$ | $19.3_{\pm1.7}$ | $25.4_{\pm1.8}$ | $33.4_{\pm2.5}$ | $21.4_{\pm1.7}$ | $33.1_{\pm3.7}$ | $17.4_{\pm2.2}$ | $27.4_{\pm1.4}$ | 18.3 | 28.3 |
| L-SBERT$_{Paraphrase}$ | - | $68.9_{\pm4.3}$ | $76.1_{\pm2.0}$ | $61.4_{\pm3.8}$ | $69.0_{\pm2.4}$ | $66.7_{\pm4.9}$ | $79.4_{\pm3.6}$ | $76.3_{\pm2.8}$ | $87.5_{\pm1.6}$ | 68.3 | 78.0 |
| L-PIE (Ours) | | $\mathbf{71.4}_{\pm4.6}$ | $\mathbf{78.2}_{\pm2.2}$ | $\mathbf{68.7}_{\pm6.1}$ | $\mathbf{76.8}_{\pm3.3}$ | $\mathbf{71.4}_{\pm5.4}$ | $\mathbf{82.6}_{\pm3.2}$ | $\mathbf{83.1}_{\pm2.4}$ | $\mathbf{90.4}_{\pm1.3}$ | **73.7** | **82.0** |
| L-BERT$_{TAPT}$ | | $65.1_{\pm4.7}$ | $73.8_{\pm2.2}$ | $53.4_{\pm4.5}$ | $64.3_{\pm1.8}$ | $63.7_{\pm4.8}$ | $75.6_{\pm2.2}$ | $73.5_{\pm3.7}$ | $82.9_{\pm1.7}$ | 63.9 | 74.2 |
| L-SBERT$_{Paraphrase}$ | ProtoNet | $\mathbf{74.9}_{\pm4.2}$ | $\mathbf{81.9}_{\pm1.9}$ | $65.2_{\pm6.0}$ | $74.4_{\pm3.2}$ | $71.8_{\pm6.6}$ | $82.9_{\pm3.5}$ | $84.3_{\pm1.4}$ | $90.9_{\pm1.0}$ | 74.0 | 82.5 |
| L-PIE (Ours) | | $74.5_{\pm4.4}$ | $81.4_{\pm2.2}$ | $\mathbf{67.2}_{\pm8.2}$ | $\mathbf{76.9}_{\pm3.9}$ | $\mathbf{73.7}_{\pm6.3}$ | $\mathbf{83.6}_{\pm3.2}$ | $\mathbf{85.4}_{\pm1.8}$ | $\mathbf{91.2}_{\pm1.2}$ | **75.2** | **83.3** |
| L-BERT$_{TAPT}$ | | $70.1_{\pm4.8}$ | $78.5_{\pm2.2}$ | $63.1_{\pm4.3}$ | $69.4_{\pm2.9}$ | $69.5_{\pm4.6}$ | $80.5_{\pm1.9}$ | $79.5_{\pm2.6}$ | $86.5_{\pm1.1}$ | 70.6 | 78.7 |
| L-SBERT$_{Paraphrase}$ | ProtAugment | $77.4_{\pm3.6}$ | $82.8_{\pm1.7}$ | $67.5_{\pm5.6}$ | $75.5_{\pm3.1}$ | $72.1_{\pm5.7}$ | $83.6_{\pm3.0}$ | $84.5_{\pm1.8}$ | $90.9_{\pm1.0}$ | 75.4 | 83.2 |
| L-PIE (Ours) | | $\mathbf{77.6}_{\pm3.4}$ | $\mathbf{82.9}_{\pm1.6}$ | $\mathbf{70.6}_{\pm6.6}$ | $\mathbf{77.4}_{\pm3.5}$ | $\mathbf{74.6}_{\pm5.7}$ | $\mathbf{84.2}_{\pm3.1}$ | $\mathbf{86.5}_{\pm1.6}$ | $\mathbf{91.8}_{\pm1.0}$ | **77.3** | **84.1** |

Table 8: N-way K-shot intent classification performance of pre-trained models with and without fine-tuning on four test set. Averaged accuracies and standard deviations across five class splits are reported. The 'L-' prefixes indicate the use of intent label names when creating prototypes. Highest scores are **boldfaced**. Fine-tuning is done in the 5-way setting due to memory constraints.[2]

**N-way K-shot intent classification** We show-case the performance of our PIE model and the baselines in a more challenging and practical scenario (Table 8). In this scenario, the intent for user utterances needs to be classified among a significantly larger number of intent classes (e.g., $10\times$ for Clinc150). The results show that our L-PIE model achieves 73.3% and 82.0% in zero- and one-shot settings, respectively, outperforming the baseline L-SBERT$_{Paraphrase}$ by 5.4% and 4.0%. These performance improvements are significantly higher than those observed in the 5-way K-shot IC task. This indicates that our PIE model performs well in practical scenarios, as stated above.

## 6 Analysis

### 6.1 Pre-training Corpus Ablation

| Pre-training Data | Average | |
|---|---|---|
| | K=0 | K=1 |
| None | 68.3 | 78.0 |
| SGD | 69.6 | 78.6 |
| DSTC11-T2 | 70.7 | 79.4 |
| TOP (+TOPv2) | 73.1 | 81.7 |
| + DSTC11-T2 | 73.3 | 81.9 |
| + SGD | **73.7** | **82.0** |

Table 9: Pre-training data ablation of L-PIE in N-way K-shot intent classification on four datasets (Banking77, HWU64, Liu54, and Clinc150). 'None' indicates L-SBERT$_{Paraphrase}$ which used as the initial encoder before applying the intent-aware pre-training method.

We leverage dialogue datasets for building the PIE model as described in Section 5.2. Here, we perform an ablation study over the pre-training datasets on N-way K-shot IC tasks (Table 9). The result shows that using the TOP (+TOPv2) dataset, which has 31K utterances, 61 gold intents, and 23K pseudo intents, improves the performance the most over L-SBERT$_{Paraphrase}$ (indicated as None in Table 9). Specifically, there is an improvement of 4.8% and 3.7% in the zero- and one-shot settings, respectively. Although using other datasets, such as SGD or DSTC11-T2, does not improve the performance in comparison with using the TOP dataset, we observe that merging them further improves the overall performance on downstream tasks.

### 6.2 Pre-training Loss Ablation

| Method | HWU64$_{N=25}$ | | Clinc150$_{N=50}$ | |
|---|---|---|---|---|
| | K=0 | K=1 | K=0 | K=1 |
| L-PIE (**Ours**) | **68.7** | **76.8** | **83.1** | **90.4** |
| - $\mathcal{L}_{pseudo}$ | 66.9 | 75.7 | 81.4 | 89.7 |
| - $\mathcal{L}_{gold\_intent}$ | **68.7** | 76.2 | 82.7 | **90.4** |
| - $\mathcal{L}_{gold\_utterance}$ | **68.7** | 75.9 | 82.7 | 90.0 |

Table 10: Loss ablation in N-way K-shot intent classification with accuracy on HWU64 and Clinc150.

As described in Section 4, our intent-aware contrastive loss comprise three sub-losses, $\mathcal{L}_{gold\_intent}$,

---

[2]When fine-tuning models in the N-way setting, we encountered out-of-memory issues on a single 12GB GPU. To work around this limitation, we fine-tuned models in the 5-way setting and evaluated them in the N-way setting.

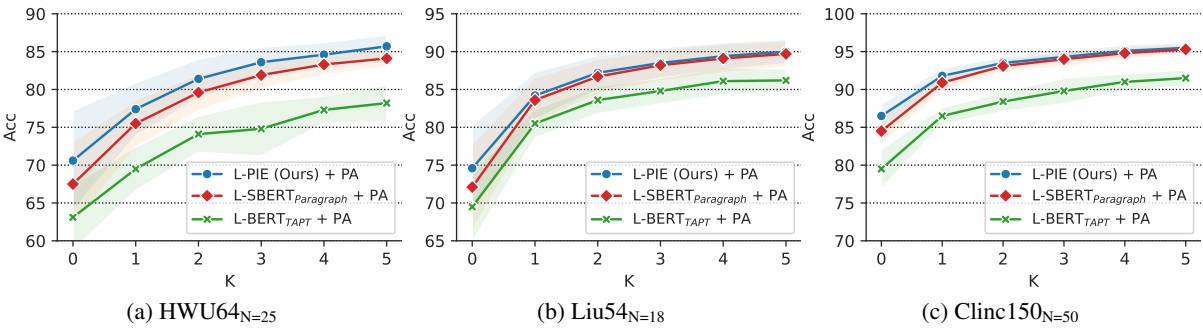

Figure 2: Performance on N-way K-shot intent classification with varying K. PA refers to ProtAugment.

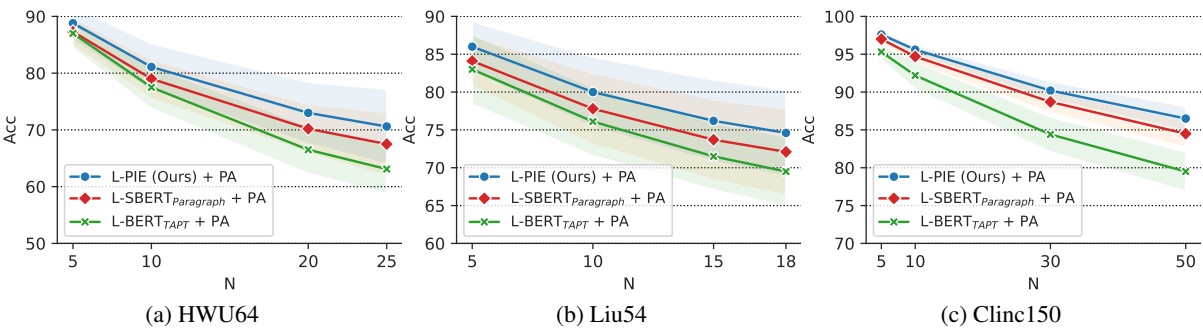

Figure 3: Performance on N-way 0-shot intent classification with varying N.

$\mathcal{L}_{\text{gold\_utterance}}$, and $\mathcal{L}_{\text{pseudo}}$. To see the benefit of using these losses during pre-training, we ablate each loss function in the N-way K-shot IC tasks on HWU64 and Clinc150 (Table 10). The results indicate that two sub-losses, $\mathcal{L}_{\text{gold\_intent}}$ and $\mathcal{L}_{\text{gold\_utterance}}$ show relatively marignal improvements. However, it is noteworthy that $\mathcal{L}_{\text{pseudo}}$ serves as the key sub-loss for PIE, highlighting the effectiveness of using pseudo intents. Specifically, removing $\mathcal{L}_{\text{pseudo}}$ from the final loss results in up to 1.8% and 1.1% degradation in performance in the zero- and one-shot settings, respectively.

## 6.3 Varying K and N

We visualize the performance of the PIE model in challenging N-way K-shot IC task settings where the number of example utterances K or the number of candidate intent classes N varies. Plots of performance at varying K (Figure 2) show that our model has consistently higher performance than the baselines, and the performance improvement of our model is the largest when K is small (e.g., K=0). Plots of performance at varying N (Figure 3) show that the performance improvement of our model increases as the number of intents N increases (i.e. increasing from N=5 to N=50). These visualizations reveal that the PIE model can be utilized in

more practical and realistic settings where many user intents are used for the TOD system and only a few utterances are available.

## 6.4 Impact of Overlapping Intents

| | Banking77$_{N=27}$ | HWU64$_{N=25}$ | Clinc150$_{N=50}$ |
|---|---|---|---|
| All intents | 71.4 | 68.7 | **83.1** |
| - overlapping intents | **72.0** | **69.6** | 82.8 |

Table 11: N-way 0-shot IC performance of L-PIE by removing overlapping intents from evaluation sets.

As shown in Table 6, there are a few overlapping intent names between pre-training data and downstream data (except for Liu54). These coincidental overlaps can hinder an accurate evaluation of the generalization ability of our model. To understand the impact of intent overlaps, we also measure the performance using only non-overlapping intents (Table 11). We observe that the impact is marginal enough to be neglected, and surprisingly, removing overlapping intents rather can lead to better performance on Banking77 and HWU64. Through further analysis, we discover that this is partly because of the bias towards pairs of utterances and intent annotated in pre-training datasets. For example, an utterance 'please play my favorite

song' in pre-training data has an intent 'play music'. Our model then incorrectly predicted 'play music' for a test utterance 'that song is my favorite', where the correct intent is 'music likeness'.

## 7 Conclusions

In this work, we propose a pre-training method that leverages pseudo intent names constructed using an IRL tagger in a semi-supervised manner, followed by intent-aware pre-training (PIE). Experiments on four intent classification datasets show that our model achieves state-of-the-art performance on all datasets, outperforming the strongest sentence encoder baseline by up to 5.4% and 4.0% in N-way zero- and one-shot settings, respectively. Our analysis shows that PIE performs robustly compared to the baselines in challenging and practical settings with a large number of classes and small number of support examples. In future work, we will explore the use of IRL and our PIE model in multi-label intent classification or out-of-scope detection tasks.

## Limitations

One limitation of our method is that while it leverages annotations from the IRL tagger, the detection of spans for certain labels, such as 'Problem,' is not accurate enough (38.1% F1 score). This is likely due to the relatively short number of annotations of this type in the training set (45 annotations). To mitigate this limitation, we could consider annotating more instances of this label or implementing techniques for handling imbalanced labels.

Another limitation is that we currently treat all IRL labels equally when constructing pseudo intents. However, the importance of each label in interpreting intent can vary. To address this, we plan to investigate treating different labels differently when pre-training the encoder (e.g. by giving more weight to 'Action' and 'Argument' labels and less weight to 'Slot' labels).

## Ethics Statement

Our proposed method is for enhancing zero- and few-shot intent classification, and it does not raise any ethical concerns. We believe that this research has valuable merits that can lead to more reliable task-oriented dialogue systems. All experiments in this study were carried out using publicly available datasets.

## Acknowledgements

We gratefully acknowledge the members of AWS AI for providing valuable feedback on the project. We would also like to thank the anonymous reviewers for their insightful comments. The part of Mujeen Sung's graduate study and, accordingly, this work was supported by National Research Foundation of Korea (NRF-2023R1A2C3004176).

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
