# OpenReview forum: "Pre-training Intent-Aware Encoders for Zero- and Few-Shot Intent Classification"
_EMNLP/2023/Conference — EMNLP 2023 Main_

### Official Review · Reviewer_QwHo · 2023-08-05

**Soundness:** 4

**Excitement:**

3: Ambivalent: It has merits (e.g., it reports state-of-the-art results, the idea is nice), but there are key weaknesses (e.g., it describes incremental work), and it can significantly benefit from another round of revision. However, I won't object to accepting it if my co-reviewers champion it.

**Missing References:**

1. Lamanov, D., Burnyshev, P., Artemova, K., Malykh, V., Bout, A. and Piontkovskaya, I., 2022, July. Template-based Approach to Zero-shot Intent Recognition. In Proceedings of the 15th International Conference on Natural Language Generation (pp. 15-28).
2. Burnyshev, P., Bout, A., Malykh, V. and Piontkovskaya, I., 2021, September. InFoBERT: Zero-Shot Approach to Natural Language Understanding Using Contextualized Word Embedding. In Proceedings of the International Conference on Recent Advances in Natural Language Processing (RANLP 2021) (pp. 208-215).

**Paper Topic And Main Contributions:**

The main idea of the paper is presentation of a novel PIE method which enables an encoder to better represent an intent.

**Reasons To Accept:**

A novel method demonstrating the improvement in representational ability of the LLMs used as encoders in task-oriented dialog task.


**Reasons To Reject:**

The proposed method seems to be pretty close to already published ones [1, 2]. It would be nice to see the comparison.
Another important point is that the authors do not compare their work with state of the art in the field.


**Reproducibility:**

4: Could mostly reproduce the results, but there may be some variation because of sample variance or minor variations in their interpretation of the protocol or method.

**Reviewer Confidence:**

4: Quite sure. I tried to check the important points carefully. It's unlikely, though conceivable, that I missed something that should affect my ratings.

---

> ### Author Rebuttal · Authors · 2023-08-29
>
> We thank the reviewer for their helpful feedback.
>
> RTR4A: Thank you for providing two relevant papers. The main difference between theirs and ours is that while the papers use templates or descriptions to enhance intent representation when fine-tuning, we employ pseudo labels constructed by an IRL tagger when pre-training. We will include these papers as related works in our final manuscript to clarify the difference. Prototypical Networks are considered the state-of-the-art method for few-shot intent classification [1]. Therefore, our focus is on comparing our work with it.\
> [1] The Devil is in the Details: On Models and Training Regimes for Few-Shot Intent Classification (Mesgar et al., 2022)

---

### Official Review · Reviewer_Tq8c · 2023-08-05

**Soundness:** 3

**Excitement:**

4: Strong: This paper deepens the understanding of some phenomenon or lowers the barriers to an existing research direction.

**Missing References:**

[1] Liu, Han, Siyang Zhao, Xiaotong Zhang, Feng Zhang, Junjie Sun, Hong Yu, and Xianchao Zhang. "A Simple Meta-learning Paradigm for Zero-shot Intent Classification with Mixture Attention Mechanism." In Proceedings of the 45th International ACM SIGIR Conference on Research and Development in Information Retrieval, pp. 2047-2052. 2022.

**Paper Topic And Main Contributions:**

This paper proposes an algorithm to generate pseudo intent names from utterances across various dialogue datasets to create a pre-training dataset. PIE (Pre-trained Intent-aware Encoder) are proposed by utilizing intent-aware contrastive learning on both gold and pseudo intent names. The model outperforms the state-of-the-art work on N-way zero- and one-shot settings on experiments on four Intent Classification datasets.

**Questions For The Authors:**

1. Do the authors consider comparing with other zero-shot intent-classification models not only using Prototypical Networks, such as [1]?
2. The expression of "pre-training" is a little bit mis-leading, since the proposed model still uses paraphrase-mpnet-base-v2 encoder as a backbone model.
3. Table 8 shows N-way settings. It is confused that "Fine-tuning is done with 5-way due to memory constraints." in caption of Table 8.

**Reasons To Accept:**

1. This paper proposes a new intent-aware contrastive learning objective to leverage gold and pseudo intents and improves intent classification performance compared to baselines.
2. The motivation of using intent role labeling to generate psudo-intents are well stated and the problems in zero-shot intent classification are identified.

**Reasons To Reject:**

1. Lack of review and investigations on intent classification and contrastive learning.
2. The effectiveness of proposed model is not strong enough as it is not demonstrated under the zero-shot settings for intent-classification model. Instead, this paper only compares to intent-classification models using Prototypical Networks.
3. the quality and amount of pre-training datasets need to be studied.

**Reproducibility:**

4: Could mostly reproduce the results, but there may be some variation because of sample variance or minor variations in their interpretation of the protocol or method.

**Reviewer Confidence:**

4: Quite sure. I tried to check the important points carefully. It's unlikely, though conceivable, that I missed something that should affect my ratings.

---

> ### Author Rebuttal · Authors · 2023-08-29
>
> We appreciate the reviewer’s careful reading, helpful feedback and suggestions!
>
> RTR3A: On lack of literature review, in addition to adding some references suggested by the reviewers, we will further enrich our final manuscript by adding more recent related work on intent classification and contrastive learning.
>
> RTR3B: On effectiveness in zero-shot setting, Table 8 demonstrates that our method outperforms the SBERT_paraphrase baseline by 5.4% in the zero-shot setting without fine-tuning. These zero-shot settings become feasible by treating intent names as prototype class representations.
>
> RTR3C: While the quality of pseudo intents may be noisy due to their automatic generation without human verification, we demonstrate that incorporating these pseudo intents during pre-training enhances the text encoder's ability to align utterances and intent class names, thereby improving zero-shot intent classification. Details regarding the size of the pre-training dataset are provided in section 5.2.
>
> Q3A:  “Did you consider comparing with other zero-shot intent-classification models not only using Prototypical Networks, such as [1]?” Prototypical Networks are a widely used and powerful method for zero- and few-shot intent classification [2, 3]. We observed that [1] also employs prototypical networks for zero-shot intent classification, as stated 'The class label description ... can be treated as the class prototype.' To provide a clearer insight, we will include experiments with other approaches, such as a cross-encoder and episodic meta-learning baseline [3], in our final manuscript.\
> [1] A Simple Meta-learning Paradigm for Zero-shot Intent Classification with Mixture Attention Mechanism. (Liu et al., SIGIR’22)\
> [2] A neural few-shot text classification reality check (Dopierre et al., EACL’21)\
> [3] The Devil is in the Details: On Models and Training Regimes for Few-Shot Intent Classification (Mesgar et al., 2022)
>
> Q3B, Q3C: Thanks for your suggestions on presentation. To prevent confusion, we will update the term to 'pre-finetuning' in the final manuscript. Initially, we called our approach 'pre-training' due to our model being trained before being fine-tuned on downstream tasks. Regarding the Table 8 caption, when we tried fine-tuning models with the N-way setting, we ran into out-of-memory issues on a single 12GB GPU. To work around this, we used a 5-way approach for fine-tuning and an N-way approach for evaluating. We'll explain this more clearly in the Implementation Details to avoid confusion in the final manuscript.

---

### Official Review · Reviewer_qFW1 · 2023-08-12

**Soundness:** 3

**Excitement:**

4: Strong: This paper deepens the understanding of some phenomenon or lowers the barriers to an existing research direction.

**Paper Topic And Main Contributions:**

This paper addresses the problem of intent classification (IC) in task-oriented dialogue systems. Specifically, the main challenge it seeks to tackle is that IC models often have difficulty generalizing when trained without a sufficient number of annotated examples for each user intent. To mitigate this issue, the paper introduces a novel pre-training method for text encoders. This method utilizes contrastive learning combined with pseudo intent labels to produce embeddings better tailored for IC tasks, thereby decreasing the necessity for extensive manual annotations. Experiments results of this paper show it has achieved the state-of-art intent classification performance on multiple benchmark datasets.

C1: The paper introduces an innovative algorithm that generates pseudo intent names from utterances gathered across multiple dialogue datasets. This addresses the problem of obtaining ample annotations for a diverse intent set.

C2: The paper proposes a pre-training strategy that leverages intent role labeling (IRL) to identify key phrases in utterances which are essential for deciphering intents. The identified phrases are then used to construct examples that pre-train a text encoder contrastively. The outcome is the creation of a model called PIE (Pre-trained Intent-aware Encoder), devised to synchronize the encodings of user utterances with their respective intent names.


**Reasons To Accept:**

S1: The idea of assigning roles to words using IRL for creating pseudo intents to pre-train the encoder is innovative and interesting.

S2: This paper has good approach presentation and detailed result analysis.

S3: This paper is well-written and has clear structure to follow.



**Reasons To Reject:**

W1: In section 3, the authors annotate the intent roles with 6 types for obtaining the IRL taggers. However, the definitions of intent roles like Action, Argument, Request, etc., may not capture all nuances across various domains or applications. Additionally, these roles may overlap in some contexts, potentially complicating the tagging process.

W2: In Table 2, while many IRL labels have high precision, recall, and F1 scores, the 'Problem' label has significantly lower scores, suggesting that this particular aspect of the model might struggle in practical applications.

W3: By formulating IRL as a sequence tagging problem, the complexity of the relationships between different labels in an utterance might be overlooked. This could lead to suboptimal intent representations.

W4: The IRL tagger uses RoBERTa-base as its foundation. While RoBERTa is a powerful model, the results might differ if another model was used. This dependency also means that improvements in RoBERTa (or discovery of issues in it) could directly affect the IRL tagger's performance. Probably an ablation study is needed for showing the IRL tagger's performance on different models and how the different IRL tagger performance may affect the IC task performance.

W5: For the intent-aware contrastive learning part, the method assumes that randomly sampled utterances that share the same gold intent names can be treated as positive pairs. This might introduce ambiguity, especially if different utterances with the same intent name have nuanced differences in meaning. Also, the method treats pairs that are not positive in a batch as negative pairs. This simplistic approach might introduce noisy negative samples, affecting CL performance.


**Reproducibility:**

4: Could mostly reproduce the results, but there may be some variation because of sample variance or minor variations in their interpretation of the protocol or method.

**Reviewer Confidence:**

4: Quite sure. I tried to check the important points carefully. It's unlikely, though conceivable, that I missed something that should affect my ratings.

---

> ### Author Rebuttal · Authors · 2023-08-29
>
> Thanks for your thorough reading and comprehensive feedback! We’re glad you found our approach innovative and interesting and that you found our presentation to be clear.
>
> RTR 2A, 2B, 2C, 2D: On IRL annotations, we acknowledge that our role labeling scheme does not capture all nuances of semantics in intent utterances. However, because our main goal is to facilitate intent encoder pre-training through generation of pseudo labels, we believe a simplified representation that captures key information in each utterance is sufficient. We intentionally adopt a small/flat (relationless) tagging process to avoid annotation complexity, even though this sacrifices some representational capacity. Of course, a more sophisticated semantic parse could facilitate more robust representations, which we plan to explore in future work.
>
> The low scores on the 'Problem' label, as discussed in the limitation section, stem from the limited problematic state utterances in the SGD dataset. Including IRL annotations on other datasets would be able to mitigate this. Despite this limitation, we still demonstrate the effectiveness of using our IRL tagger for constructing pseudo labels on zero- and few-shot intent classification datasets that include more utterances with problem states (like Banking 77).
>
> Thanks for the suggestion to conduct further ablation studies to understand the impact of IRL tagger performance on downstream tasks. Though we agree this may lead to further useful insights, we selected RoBERTa over other encoders (BERT-base) based on IRL tagger validation set performance. Our focus in this work is on comparison of pre-training with or without IRL tags.
>
> RTR2E: We acknowledge that different datasets may have different granularities for defining intents (e.g. one FileClaim intent vs. multiple fine-grained FileAutoClaim / FilePetClaim intents) – understanding the effect of this potential noise in contrastive learning specific to intent encoder pre-training is left for future work.

---

### Official Review · Reviewer_wRCq · 2023-08-12

**Soundness:** 4

**Excitement:**

4: Strong: This paper deepens the understanding of some phenomenon or lowers the barriers to an existing research direction.

**Paper Topic And Main Contributions:**

The authors introduce a unique pre-training approach for text encoders, leveraging contrastive learning combined with intent pseudo-labels. The primary motivation behind this methodology is to produce embeddings highly conducive to Intent Classification (IC) tasks, potentially minimizing the dependency on manual annotations. The proposed method is called PIE (Pre-trained Intent-aware Encoder). The PIE model's design intends to synchronize the encodings of utterances with their associated intent designations. The procedure is detailed and appears to be executed in two primary stages: 1. Initial training of a tagger that pinpoints pivotal phrases within utterances, which are deemed essential for discerning intents. 2. Utilization of these identified phrases as instances for the pre-training of a text encoder, executed in a contrastive fashion. As per the experiments demonstrated, there is notatble improvement in accuracy with the PIE model recording a 5.4% and 4.0% augmentation in performance when compared to the erstwhile state-of-the-art pre-trained sentence encoder. This observation is consistent for the N-way zero- and one-shot settings evaluated across four distinct IC datasets.

**Questions For The Authors:**

Is there any ways that the author could share the details of the annotated IRL datasets to help community to boostrap a tagger.

**Reasons To Accept:**

The paper proposed a novel and inspiring method for the IC task and achieve STOA performance. The method can potentially be useful for other NLP tasks as well.

**Reasons To Reject:**

No obvious flaws.

**Reproducibility:**

4: Could mostly reproduce the results, but there may be some variation because of sample variance or minor variations in their interpretation of the protocol or method.

**Reviewer Confidence:**

3: Pretty sure, but there's a chance I missed something. Although I have a good feel for this area in general, I did not carefully check the paper's details, e.g., the math, experimental design, or novelty.

---

> ### Author Rebuttal · Authors · 2023-08-29
>
> Thanks for your careful reading, encouraging feedback and interest in our method and dataset.
>
> Q1: “Is there any way that the author could share the details of the annotated IRL datasets to help the community to bootstrap a tagger?” We will publicly release our IRL annotations to help readers gain a clearer understanding of the annotation descriptions outlined in the paper.

---

### Meta-Review · Area_Chair_q18J · 2023-09-19

**Recommendation:** 4

**Metareview:**

Overall, the reviewers are positive about the paper, highlighting its novelty, innovative approach, and strong performance in intent classification. However, Reviewer 2 raises some concerns about the intent role labeling, the use of the RoBERTa-base model, and the treatment of positive and negative pairs in the contrastive learning. Reviewer 3 also mentions the lack of investigation on intent classification and contrastive learning and suggests comparing with other zero-shot intent-classification models. Reviewer 4 appreciates the novelty of the method but suggests comparing it with similar published work and the state-of-the-art.

In terms of pros, the paper is commended for proposing a unique pre-training approach for text encoders, leveraging contrastive learning and intent pseudo-labels. The method is considered inspiring and has the potential for other NLP tasks. The motivation behind the methodology is clearly stated, and the experiments show significant improvement in performance over the state-of-the-art pre-trained sentence encoder. Additionally, the paper is well-written, has a clear structure, and provides sufficient support for its claims.

In terms of cons, the concerns raised by Reviewer 2 regarding intent role labeling, the RoBERTa-base model dependency, and the treatment of positive and negative pairs in contrastive learning should be addressed. Reviewer 3 suggests conducting further investigation and comparing with other zero-shot intent-classification models. Reviewer 4 suggests comparing the proposed method with similar published work and the state-of-the-art.

---

### Decision · Program_Chairs · 2023-10-07

**Decision:**

Accept-Main

**Comment:**

Overall, the reviewers are positive about the paper, highlighting its novelty, innovative approach, and strong performance in intent classification. However, Reviewer 2 raises some concerns about the intent role labeling, the use of the RoBERTa-base model, and the treatment of positive and negative pairs in the contrastive learning. Reviewer 3 also mentions the lack of investigation on intent classification and contrastive learning and suggests comparing with other zero-shot intent-classification models. Reviewer 4 appreciates the novelty of the method but suggests comparing it with similar published work and the state-of-the-art.

In terms of pros, the paper is commended for proposing a unique pre-training approach for text encoders, leveraging contrastive learning and intent pseudo-labels. The method is considered inspiring and has the potential for other NLP tasks. The motivation behind the methodology is clearly stated, and the experiments show significant improvement in performance over the state-of-the-art pre-trained sentence encoder. Additionally, the paper is well-written, has a clear structure, and provides sufficient support for its claims.

In terms of cons, the concerns raised by Reviewer 2 regarding intent role labeling, the RoBERTa-base model dependency, and the treatment of positive and negative pairs in contrastive learning should be addressed. Reviewer 3 suggests conducting further investigation and comparing with other zero-shot intent-classification models. Reviewer 4 suggests comparing the proposed method with similar published work and the state-of-the-art.